# Transcriptome Analysis of Alcohol Drinking in Non-Dependent and Dependent Mice Following Repeated Cycles of Forced Swim Stress Exposure

**DOI:** 10.3390/brainsci10050275

**Published:** 2020-05-02

**Authors:** Sean P. Farris, Gayatri R. Tiwari, Olga Ponomareva, Marcelo F. Lopez, R. Dayne Mayfield, Howard C. Becker

**Affiliations:** 1Department of Anesthesiology and Perioperative Medicine, University of Pittsburgh, Pittsburgh, PA 15261, USA; farrissp@pitt.edu; 2Department of Biomedical Informatics, University of Pittsburgh, Pittsburgh, PA 15206, USA; 3Waggoner Center for Alcohol and Addiction Research, The University of Texas at Austin, Austin, TX 78712, USA; gayatri281@austin.utexas.edu (G.R.T.); olgaponomar@austin.utexas.edu (O.P.); dayne.mayfield@austin.utexas.edu (R.D.M.); 4Charleston Alcohol Research Center, Department of Psychiatry and Behavioral Sciences, Medical University of South Carolina, Charleston, SC 28425, USA; lopezm@musc.edu; 5Department of Neuroscience, The University of Texas at Austin, Austin, TX 78712, USA; 6Department of Neuroscience, Medical University of South, Charleston, SC 29425, USA; 7Department of Veterans Affairs Medical Center, Charleston, SC 20401, USA

**Keywords:** alcohol drinking, dependence, stress, RNA-Sequencing, prefrontal cortex, mouse

## Abstract

Chronic stress is a known contributing factor to the development of drug and alcohol addiction. Animal models have previously shown that repeated forced swim stress promotes escalated alcohol consumption in dependent animals. To investigate the underlying molecular adaptations associated with stress and chronic alcohol exposure, RNA-sequencing and bioinformatics analyses were conducted on the prefrontal cortex (CTX) of male C57BL/6J mice that were behaviorally tested for either non-dependent alcohol consumption (CTL), chronic intermittent ethanol (CIE) vapor dependent alcohol consumption, repeated bouts of forced swim stress alone (FSS), and chronic intermittent ethanol with forced swim stress (CIE + FSS). Brain tissue from each group was collected at 0-h, 72-h, and 168-h following the final test to determine long-lasting molecular changes associated with maladaptive behavior. Our results demonstrate unique temporal patterns and persistent changes in coordinately regulated gene expression systems with respect to the tested behavioral group. For example, increased expression of genes involved in “transmitter-gated ion channel activity” was only determined for CIE + FSS. Overall, our results provide a summary of transcriptomic adaptations across time within the CTX that are relevant to understanding the neurobiology of chronic alcohol exposure and stress.

## 1. Introduction

The etiology of neuropsychiatric disorders is multifaceted, characterized by the continual interaction of biological and environmental forces. Stress is a major pervasive source of environmental pressure influencing biological control of mental and emotional well-being. Maladaptive responses to acute and chronic stress can facilitate persistent allostatic mood states of behavior (the physical and psychological toll paid to maintain stability in the face of stress exposure). Such stress-induced imbalances in positive- and negative-reinforcement of behavioral control contributes to the development of substance use disorders, including alcohol use disorder (AUD) [1]. Studying the neurobiological adaptations that occur in relation to stress and substance use (e.g., alcohol consumption) is important for understanding the pathogenesis of individual neuropsychiatric disorders and comorbid disease phenotypes. 

Transcriptional reprogramming of gene expression within the central nervous system (CNS) has been proposed as a fundamental molecular process underlying repeated exposure to substances of abuse [2]. Examination of human postmortem brain tissue and animal models for AUD have demonstrated biologically coordinated changes in gene expression throughout several cell-types and areas of the CNS. Specific groups of evolutionarily conserved protein-coding and non-coding gene expression networks are associated with lifetime consumption of alcohol [3,4], suggesting chronic neuroadaptations proportional to substance use. Despite the valuable contributions of postmortem human brain to understanding the neurobiological impact of long-term alcohol use [5], not all of the psychological and pathophysiological measures are appropriately measured or rigorously controlled in human samples. Animal models provide essential translational tools for investigating the direct effects of alcohol and interaction with environmental variables, such as repeated stress, withdrawal, and periods of abstinence.

The rodent model of repeated forced swim stress (FSS) is a chronic test of inescapable stress, inducing a negative state of coping behavior, a prolonged period of poor performance due to uncontrollable and adverse conditions [6,7]. In a mouse (C57BL/6J) model of chronic intermittent ethanol (CIE) vapor exposure, known to affect tolerance and increase ethanol consumption [8,9], FSS causes robust increases of ethanol consumption in mice with a history of CIE exposure [10,11]. The added component of FSS in CIE mice is associated with greater amounts of alcohol consumption than either a no stress exposure or a model of repeated social defeat stress and does not affect more moderate levels of alcohol consumption in non-dependent mice [10]. Escalation of alcohol consumption in dependent mice via the interaction of FSS suggests, at least in part, convergent biological pathways activated by chronic stress and alcohol [11].

In an effort to better understand the shared and nonshared molecular adaptations in response to FSS and CIE, the current investigation examined transcriptome-wide alterations in gene expression across the medial prefrontal cortex (CTX), a brain-region known to be involved in decision-making and compulsive alcohol consumption [12]. The combined effect of FSS and CIE on changes in global gene expression that occur within the CTX is currently unknown. Prior studies have demonstrated brain-regional and temporal changes in gene expression for male C57BL/6J mice with a history of CIE [13,14], providing supportive evidence for widespread CNS plasticity following chronic alcohol exposure on the CTX and other brain-regions. Our results demonstrate conditional long-term changes in CTX gene expression for behavioral models of alcohol consumption and chronic inescapable stress. The combined effect of this measure of stress and alcohol consumption is essential for continued adaptations in CNS plasticity.

## 2. Materials and Methods

*Animal Model:* Adult male C57BL/6J (B6J) mice, 10 weeks of age, were purchased from Jackson Laboratories (Bar Harbor, ME) and individually housed under a 12-h light/dark cycle (lights on at 8:00 a.m.). Animals were given ad libitum access to food (Teklad rodent diet) and water in a temperature and humidity controlled AAALAC-accredited facility at the Medical University of South Carolina. All procedures were approved by the Institutional Animal Care and Use Committee and conducted in accordance with the NIH Guide for the Care and Use of Laboratory Animals.

*Procedure:* Mice were evaluated in the CIE-FSS Drinking model, as illustrated in Figure 1. Briefly, mice were first trained to drink 15% (*v*/*v*) ethanol with access for 1 h/day starting 3 h after lights off. Once stable baseline intake was established (~3 weeks), mice were separated into four groups (equated for baseline level of alcohol intake): non-dependent alcohol consumption (CTL), FSS, CIE, and CIE + FSS. Mice in the CIE and CIE+FSS groups received CIE vapor exposure in inhalation chambers (16 h/day for 4 days), as detailed below. Mice in the remaining groups (CTL and FSS) were similarly handled but maintained in air inhalation chambers. As in our previous work [10,11], all groups received pyrazole (1 mmol/kg) injections (ip) immediately prior to being placed into inhalation chambers (to stabilize BEC in CIE and CIE + FSS groups). Limited access drinking sessions were suspended during inhalation exposure. After a 72-h abstinence period, drinking test sessions resumed for 5 consecutive days under the same limited access conditions as before. This pattern of weekly chronic intermittent ethanol (or air) exposure alternated with weekly limited access drinking sessions was repeated for 4 cycles (Test Cycles 1–4) (Figure 1). Mice in the FSS and CIE + FSS groups experienced 10 min of forced swim stress (FSS) 4 h prior to each of the test drinking sessions, as detailed below. The remaining mice (CTL and CIE groups) remained in their home cage undisturbed. As indicated in Figure 1, mice were sacrificed after a fifth and final CIE (or air control) cycle, either immediately (0-h), 72 h, or 7 days after the last CIE (or air) exposure. Mice were decapitated, and whole brains were immediately snap-frozen.

*CIE Exposure:* Mice in the CIE and CIE + FSS groups received chronic intermittent ethanol (CIE) vapor exposure in inhalation chambers (16 h/day for 4 days), while CTL and FSS mice were similarly handled but exposed to air in control chambers. CIE exposure was administered in inhalation chambers according to procedures previously described [8,9,10]. Briefly, mice were placed in Plexiglas inhalation chambers (60 W × 60 L × 36 H cm) and exposed to ethanol vapor at levels set to yield stable blood ethanol concentrations (BEC) in C57BL/6J mice in the range of 175–225 mg/dL. Separate sets of inhalation chambers were used for animals to better control vapor concentrations that yielded similar BEC. Housing conditions were similar to those in the colony room. Blood samples (40 µL) were obtained from the retro-orbital sinus with heparinized capillary tubes during each CIE cycle of exposure. Blood samples were centrifuged, and plasma was processed in an Analox Instrument analyzer (Analox Instruments USA, Lunenburg, MA, USA), with blood ethanol levels expressed as mg/dL. 

*Forced Swim:* Mice were placed in a glass cylinder (20-cm diameter × 40-cm tall) filled with 23–25 °C water for 10 min [10,11].

*Design and Data Analysis:* Voluntary ethanol intake in grams per kilogram of body weight was the main behavioral dependent variable under analysis. Daily intake values were averaged over the last five days of baseline and the five days of each test cycle. Less than 1% of the data points obtained were aberrant due to leaks. Those values were replaced by the weekly mean value for that particular subject. Data were analyzed by ANOVA, with group (CTL, FSS, CIE, or CIE + FSS) and time of brain collection (0-h, 72-h, or 7 days) as between-subject factors and phase (baseline, Tests 1–4) as a repeated measure (*n* = 7–10/group). BECs were also analyzed by ANOVA, with group and exposure cycle as the main factors. For both analyses, significant main effects and interactions were further analyzed using the Newman-Keuls test for paired comparisons. Statistica 13 (TIBCO Software Inc., Palo Alto, CA, USA) was used for these data analyses. The experimental design is illustrated in Figure 1.

*Brain Tissue Dissection and RNA Isolation:* Frozen brains were embedded in a plastic mold containing ice-cold optimal cutting temperature compound (OCT) and quickly frozen in an isopentane/powdered dry ice system. A Microm HM550 cryostat (Thermo Scientific, Ontario, CA, USA) was used for sectioning at a thickness of 300 μm. Micropunches were bilaterally collected from mouse prefrontal cortex (CTX 1.5 mm; Bregma 2.96mm–1.98mm) and individually pooled per mouse for total RNA extraction at each of the timepoints and from a naive subjects group. Total RNA was isolated using the MagMAX-96 Total RNA Isolation kit (#1839 Thermo Fisher Scientific Inc., Rockford, IL, USA) and DNAse-treated according to the manufacturer’s instructions (#1906, Thermo Fisher Scientific Inc., Rockford, IL, USA). Quality and quantity of isolated total RNAs were characterized with a nanodrop 8000 spectrophotometer (Thermo Fisher Scientific, Rockford, IL, USA; 260/280 and 260/230 ratios), Agilent 2200 TapeStation Instrument (Agilent Technologies, Santa Clara, CA, USA; RNA-integrity numbers ranged between 9–10), and Qubit fluorometric quantitation (Thermo Fisher Scientific). An aliquot of 400-ng total RNA (based on Qubit) was used for RNA-sequencing.

*RNA-Sequencing and Bioinformatics Analyses:* Isolated total RNA from mouse CTX was submitted to the Genomic Sequencing and Analysis Facility at The University of Texas at Austin. Sequencing libraries were constructed using a 3’ Tag-based approach (TagSeq), targeting the 3’ end of RNA fragments [15] from ~16 ng/µL of each RNA sample. This tag-based approach is a cost-efficient alternative to whole-transcriptome RNA sequencing, comparable with respect to accuracy and quantification of detected transcripts [16]. Samples were sequenced on the HiSeq 2500 (Illumina) platform with a read depth of approximately 7.6 million reads (single-end 100 bp reads; 4.6 million high quality reads per sample after trimming). A total of 95 samples were included with an average of 7.6 million reads per sample. There were 3.1 million uniquely mapped reads after mapping (91% mapping). RSeQC read distribution identified ~ 980K 3’UTR exon reads that were 32% of uniquely mapped reads. TagSeq detected a total of 56,350 transcripts. On average, 38,000 transcripts per sample were detected, representing ~22,000 protein coding genes. Sequencing sample size (n) for each group: CIE+FSS: 000h (*n* = 7), 072h (*n* = 6), and 168h (*n* = 7); CIE: 000h (*n* = 8), 072h (*n* = 5), and 168h (*n* = 9); FSS: 000h (*n* = 9), 072h (*n* = 7), and 168h (*n* = 10); and CTL: 000h (*n* = 9), 072h (*n* = 8), and 168h (*n* = 10). Duplicated reads (5’) sharing the same degenerate header and the first 20 bases of the sequence was trimmed off fastq files. Read quality was assessed using MultiQC (version 1.7). Reads were mapped to the mouse reference genome (Gencode GRCm38.p6 release M22) using a STAR (version STAR_2.5.4b) aligner [17]. Read distribution of bam files over reference genomic features was evaluated with RSeQC (version 3.0.0) [18]. Raw counts were quantified using HTSeq (version 0.11.2). ERCC RNA Spike-In Mix 1 (Thermo Fisher Scientific, Rockford, IL, USA; Part no 4456740) was used to assess platform dynamic range and lower limit detection. 

*Differential Gene Expression and Time-Course Cluster Analysis:* The R (version 3.5.1) package DESeq2 (version 1.22.2) [19] was used to identify changes in expression across different levels with a likelihood ratio test (LRT) using the *DESeq* function. To identify changes in gene expression from a common baseline, individual groups and timepoints were initially compared against naive subjects. A nominal *p*-value less than or equal to 0.05 was selected to ascertain shared and nonshared changes in gene expression among the groups and timepoints with respect to affected biological categories and pathways. The full DESeq model of *~Group + Time + Group*:*Time,* versus the reduced model *~Group + Time,* was used to identify transcriptome patterns using the clustering tool *degPatterns* from the “DEGreport” R package (version 1.19.1). The function was run using the default parameters, except that the produced clusters were allowed to contain a minimum of one gene representative (minc = 1). This produced clusters (*n* = 14) based on similar profiles. In order to establish a set of stringent gene expression clusters according to both group and time, statistically significant changes were limited to genes less than or equal to a false discovery rate of 5%.

*Gene Ontology, Molecular Pathway, and Cell-Type Analysis:* Differentially expressed genes were analyzed for enrichment of canonical gene ontologies and molecular pathways using the bioinformatic tool Enrichr [20,21]. Reported ontological categories and pathways were limited to those consistently altered across all timepoints within each of the separate experimental groups (*p* < 0.05). Differentially expressed genes (*Group + Time*) were compared to genes with 10-fold higher expression in either murine neurons, oligodendrocytes, astrocytes, and microglia [22] to determine the overrepresentation of the major CNS cell-types using a Fisher’s exact test (*p* < 0.05).

## 3. Results and Discussion

### 3.1. Blood Ethanol Concentration during CIE Exposure Cycles

Mice were exposed to ethanol vapor concentrations set to produce BEC around 200 mg/dL. ANOVA indicated a significant group × cycle interaction (*F*(4,188) = 2.75; *p* < 0.05), and post-hoc comparisons indicated that mice in the CIE + FSS mice exhibited higher BEC than CIE mice during Cycles 2 and 3 of intermittent ethanol vapor exposure (Figure 2). Importantly, blood ethanol levels were similar between CIE and CIE + FSS groups during the last two exposure cycles (Cycles 4 and 5) before brain samples were collected (Figure 2). Additionally, BECs were similar between CIE and CIE+FSS groups at the time brains were collected immediately after the last 16-hr vapor exposure (*F*(1,15) = 0.28; *p* = NS) (Figure 2).

### 3.2. Voluntary Ethanol Intake

The ANOVA indicated significant main effects of group (*F*(3,97) = 64.07; *p* < 0.0001) and phase (*F*(4,388) = 70.82; *p* < 0.001) and a significant interaction between these factors (*F*(3,388) = 23.05; *p* < 0.0001). Importantly, the time of brain collection did not have a significant effect (*F*(2,97) = 0.24) and did not have a significant interaction with the other factors under analysis. Post-hoc comparisons based on the significant group × phase interaction indicated that mice in the CIE and CIE + FSS groups consumed more ethanol during all test cycles compared to CTL mice within each test cycle and their own baseline (* in Figure 3). This analysis also showed that mice in the FSS group had a lower intake level during Test 3 compared to their own baseline (^ in Figure 3). Importantly, mice in the CIE + FSS evidenced significantly higher levels of intake compared to CIE mice during test Cycles 2, 3, and 4 (# in Figure 3).

### 3.3. Overview of Differential Gene Expression 

Stress and addiction to alcohol, as well as other substances of abuse, are interlinked components of mental health. Identifying the shared and unshared biological systems impacted for these conditions represents an important step in understanding the pathophysiology of diseases associated with AUD and stress. The current studies sought to test the hypothesis that nondependent alcohol consumption (CTL), chronic intermittent ethanol (CIE) vapor-induced dependence, repeated forced swim stress (FSS), and the combination of CIE and FSS produce distinct transcriptomic signatures within the prefrontal cortex, a critical brain-region involved in stress and alcohol addiction [23,24]. RNA-seq analysis was conducted on total RNA isolated from the medial prefrontal cortex (referred to as CTX throughout the manuscript) to identify transcriptome-wide gene expression changes in four treatment groups compared to naive subjects: CTL, CIE, FSS, and CIE + FSS. In order to characterize temporal patterns of transcriptomic responses underlying the allostatic load for each behavioral condition, cortical tissue was collected at 0 h, 72 h, and 168 h following the final experimental session (Figure 1). Investigating the transcriptome response of multiple timepoints is required for determining early, intermediate, and long-lasting molecular adaptations that occur in animal models of human disease. The biological contribution from each of these timepoints may be important mediators in the development and continuation of abnormal behavioral responses. 

We identified 1651–4322 differentially expressed genes with nominal *p*-values less than or equal to 0.05. These changes vary in a Cohen’s d effect size between −6.57 and 7.65, with a mean absolute effect size of 1.56. In general, the number of significant expression changes decreased as a function of time (0 h > 72 h > 168 h; Figure 4A). Across all treatment groups and timepoints, there was a greater number of upregulated compared to downregulated genes. Each of the separate treatment groups exhibited mostly unique transcriptome signatures across each of the examined timepoints (Figure 4B). 

Time-dependent patterns of gene expression changes shared within each treatment group ranged from 24% (FSS 0 h & 168 h) to 57% (FSS 0 h & 72 h). This degree of overlapping changes was generally stronger for increased versus decreased gene expression (Figure 5). Although many alterations in gene expression were unique across treatment groups and timepoints, the considerable overlap ranging from 204–488 genes may suggest that opportunity to voluntarily drink ethanol (albeit at different levels in each of the independent treatment groups) induced unique long-lasting alterations to the CTX transcriptome. However, it is also possible that compared to naïve subjects, the degree of overlapping changes in the gene expression may be partially due to the alcohol dehydrogenase inhibitor pyrazole. In concordance with our previous microarray study from the CTX on time-dependent (0-h, 8-h, and 120-h) gene expression changes due to repeated chronic intermittent ethanol (CIE) vapor exposure [13], there were 1800 overlapping genes also identified in the current time-course. As a result of the technical advantages of RNA-seq compared to microarrays, as well as a different time trajectory collected here, the current study was powered to detect up to 4322 differentially expressed genes for CIE. Despite such differences, these studies strongly suggest persistent coordinately regulated changes to the transcriptome as a result of chronic alcohol exposure.

Similar to previous studies on time-dependent changes in gene expression [13,14], our results demonstrate short- and long-term effects on coordinately regulated biological systems by CIE (Appendix A). For example, brain-derived neurotrophic factor (*Bdnf*), which in the present study has decreased expression in the CTX of CIE mice at 0 h and 72 h, is an important nerve growth factor involved in CNS plasticity and escalation of alcohol consumption [25,26]. Increased expression of *Bdnf* in the prefrontal cortex of adult male C57BL/6J mice reduces CIE-induced alcohol consumption [27]. CIE decreases *Bdnf* expression in the medial prefrontal cortex of adult male C57BL/6J mice and increases the expression of microRNA miR-206 [28]. This microRNA selectively suppresses *Bdnf* expression in the medial prefrontal cortex of alcohol-dependent rats [29]. Our analysis shows CIE dependent gene expression is consistently overrepresented for miR-206 gene targets at the 0-h, 72-h, and 168-h timepoints (Appendix A). None of the other tested behavioral groups (i.e., CTL, FSS, and CIE + FSS) consistently altered the expression of known miR-206 targets across all of the examined timepoints. CIE may be an experimental factor involved in the regulation of the expression of specific noncoding (e.g., miR-206) and protein-coding transcripts controlling distinct aspects of excessive alcohol consumption and addictive behavior. Additionally, the current study demonstrated shared, and nonshared, changes in gene expression for alcohol dependence and withdrawal (CIE) with non-dependent alcohol consumption and forced swim stress (Figure 5).

Chronic uncontrollable stress is a significant source of physical and psychological strain. Genotype differences between different strains of mice influences measures of stress and adaptive behavior, with C57BL/6J mice demonstrating greater baseline immobility and the least amount of experimental variation in the forced swim test among 11 tested strains of mice [25]. Using the repeated forced swim stress (FSS) mouse behavior model, our transcriptome analysis indicated this stress exposure produces an immediate and prolonged increase in genes related to steroid-related transcription and epigenetic control of gene expression (Table 1). Dysregulation of the HPA-axis due to environmental stressors disrupts the proper secretion of corticosteroids, causing an imbalance in signaling processes and structural changes in several brain regions, including the prefrontal cortex [26]. The prefrontal cortex is an essential component of the neurocircuitry involved in decision-making and providing negative feedback to the stress-evoked activation of the HPA-axis [27,28,29]. Aberrant levels of corticosteroids are capable of distorting dopaminergic function in mesocortical neurons through the alteration of DNA methylation [30]. Inhibition of dopaminergic neurons in mice adversely affects the motivated behavior using the forced swim test without affecting the generalized locomotor activity [31]. Repeated FSS exposure could establish a physiological state of anhedonia, which when combined with CIE, gives rise to a selective upregulation in expression of genes involved in the “response to dopamine” (GO:1903350; Appendix A).

The Bioconductor package DEGreport includes a *degPatterns* function that uses a hierarchical clustering approach based on pairwise correlations to identify sets of genes with consistent patterns of expression as a function of both time and treatment group (Appendix A). We used degPatterns to identify distinct temporal patterns of expression across treatment groups. The list of 292 statistically significant genes (padj < 0.05) identified by DESeq2 was used as input for the analysis. A total of 14 clusters (distinct expression patterns) were identified. The number of genes per cluster ranged from 2–70 genes per cluster (Figure 6). The largest cluster (Cluster 1 of Figure 6 and Appendix A) shows a transient increase at 0 h followed by a stepwise decrease in the expression of genes with time following CIE or CIE + FSS treatments. The neuro-immuno-responsive interferon genes *Irf7*, *Ifit1, Ifit3,* and *Ifitm3* represented in Cluster 1 are temporarily increased in the CTX, gradually returning to baseline levels. This is accompanied by the same pattern of expression change in other genes such as the enzyme *Plat* and neuropeptide *Vip*. Vasoactive intestinal peptide (Vip) is known to selectively activate interferon alpha/beta synthesis in glial cells [32], suggesting a potential short-term modulatory role of the neuroimmune system following CIE. Cluster 3 (Figure 6 and Appendix A) shows a similar trend represents genes like *Ctss* and *B2m*, which were identified as novel genetic determinants of complex behavioral traits and neuroimmune signaling in the regulation of alcohol consumption [33]. Another prominent pattern of expression includes genes that increase as a function of time in CIE and CIE + FSS groups. For example, Cluster 7 (Figure 6 and Appendix A) includes the alcohol responsive genes *Pde4a* and *Gabbr2*. Phosphodiesterases regulate immune pathways, and PDE4, in particular, has been implicated in alcohol and substance use disorders. The PDE4 inhibitor rolipram has been shown to reduce ethanol intake and preference in mice [34], and apremilast produces stable decreases in voluntary ethanol consumption [35]. Additionally, preclinical and clinical data suggest a therapeutic potential for ibudilast, a more general PDE inhibitor [36,37]. In contrast, Cluster 10 (Figure 6) shows a striking stepwise decrease in the expression of 8 genes: *Cbln4, Cdhr1, Rab37, Calb2, Traf7, Doc2g, Eomes,* and *Cers2* across timepoints only in the CIE + FSS treatment. These genes are unchanged in clusters from the other treatment groups. *Cers2* shows enrichment in molecular pathways and gene ontogenies for sphingolipid metabolism. The sphingolipid system has been implicated as a molecular mechanism involved in alcohol’s self-medicating effects on innate depression [38]. In addition, *Cers2* is known to be involved in the regulation of axon regeneration, which may be important after long-term withdrawal from alcohol. Together, these novel results implicate specific genes associated with combined alcohol dependence and stress.

Neuropsychiatric disorders are recurrent negative states of human well-being affected by genetic predisposition and environmental factors. Repeated exposure to stressful and harmful environmental conditions interacts with the genetic factors, shaping neurobiological pathways and subsequent behavior. Using gene expression measurements as a surrogate for biological systems affected by CTL, CIE, FSS, and CIE + FSS, the current study suggests multiple systems are affected by chronic ethanol, chronic stress, and the combination of both experiences (Appendix A). The identified affected biological systems were filtered for recurring changes at each of the individual timepoints but which were also selectively observed in only a single behavioral group (CTL, CIE, FSS, and CIE + FSS). Canonical molecular-level activity for increased gene expression (Table 1A) demonstrated one unique result for CIE and CIE+FSS: phosphatidylinositol binding (GO:0035091) and transmitter-gated ion channel activity (GO:0022824), respectively. Phosphatidylinositol-binding sites, including the sorting nexin family (e.g., *SNX9*, *SNX13*, and *SNX17*), are important for facilitating intracellular trafficking and cellular signaling. Transmitter-gated ion channels, such as genes encoding for GABA-A receptors (*GABRA3*) and glutamate receptors (*GRIA1, GRIK1,* and *GRIK2*), are known molecular sites of interaction for alcohol [39]; however, persistent gene expression of these ion channels appears dependent upon the added component of repeated stress. This lasting expression of such ion channels due to FSS may involve the regulation of nuclear receptor binding (e.g., steroid hormone receptor binding (GO:0035258)) and epigenetic control (e.g., histone methyltransferase activity (GO:0042054)). Each of the treatment groups showed consistent evidence for a more limited number of molecular functions (Table 1B). The coupling of CIE and FSS uniquely downregulates genes involved in syntaxin binding (GO:0019905) and ubiquitin-protein transferase activity GO:0004842). Attenuated expression of syntaxin-binding genes (e.g., *SYT4, SYT11, RAB11A,* and *NAPA*) may be a compensatory mechanism for managing the release of neurotransmitters targeting increased expression of critical ion-channels.

The mammalian central nervous system is a heterogeneous mixture of cell types, including neurons, oligodendrocytes, astrocytes, and microglia. Each of these CNS cell types expresses a compendium of genes, characteristic of their biological function for responding to environmental stimuli and regulating behavior. To identify the cellular systems affected by each of the behavioral treatment groups (CTL, CIE, FSS, and CIE + FSS), we determined if any of the main CNS cell types were significantly enriched for differentially expressed genes from each condition (Table 2). With the exception of the nondependent alcohol drinking, neuronal expression signatures were enriched in both up- and downregulated genes across treatment groups and timepoints. Microglia- and astrocyte-related gene expression were increased for only CIE and CIE + FSS animals at the 0-hr timepoint, which may suggest chronic intermittent ethanol vapor exposure induces a generalized inflammatory state that dissipates over time. At the 72-h timepoint, all behavioral groups were overrepresented for increased gene expression in both neurons and astrocytes. Although only a modest number of genes were increased for astrocyte-related expression at this timepoint, this may indicate a temporal homeostatic relationship between astrocytes and the neuronal function. Epigenetic and transcriptional reprogramming of cellular responses is a latent feature of addiction to alcohol and other abused substances [40,41]. As previously mentioned, CIE+FSS was the only condition to consistently elicit increased expression of genes with recognized roles in “transmitter-gated ion channel activity” (GO:0022824). This may suggest a concerted effect of CIE+FSS (i.e., dependence and stress) on receptor-mediated plasticity in the CTX, mediating the chronic escalation of alcohol consumption observed in this animal model. The current studies suggest a unique set of genes distinguishing biological differences within the CTX between non-dependent alcohol consumption (CTL), CIE, FSS, and CIE + FSS.

## 4. Conclusions

Repeated and uncontrolled behavioral disturbances are a significant source of environmental pressure capable of adversely impacting mental health. Using an animal model of chronic stress and alcohol dependence, our current studies demonstrate an array of transcriptomic alterations within the CTX that relate to increased alcohol consumption associated with chronic alcohol (CIE) exposure, as well as the capacity of stress (FSS) to further enhance elevated drinking in dependent mice. Assessing time-dependent changes in gene expression reveals both transitory periods of transcriptional regulation, as well as persistent (long-lasting) gene expression changes associated with this CIE-stress drinking model. Through a comprehensive evaluation of transcriptome-wide disruptions in the regulation of gene expression, results from the present study provide a useful resource for discovering unknown biological contributors to stress-related excessive alcohol drinking associated with dependence.

## Figures and Tables

**Figure 1 brainsci-10-00275-f001:**
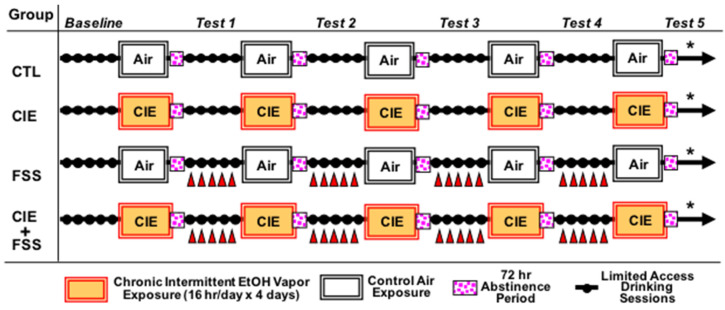
Experimental design for the chronic intermittent ethanol–forced swim stress (CIE-FSS) drinking model. The study design yielded 4 experimental groups: non-dependent alcohol consumption (CTL), CIE, FSS, and CIE + FSS. The “*” represents the time of sacrifice (0-h or 72-h or 168-h) after the last CIE/Air exposure. Only the last 5 days of baseline are represented in this scheme, and the red triangles indicate stress (FSS) exposure. A separate group of naïve mice were sacrificed and used as a common reference for gene expression analysis.

**Figure 2 brainsci-10-00275-f002:**
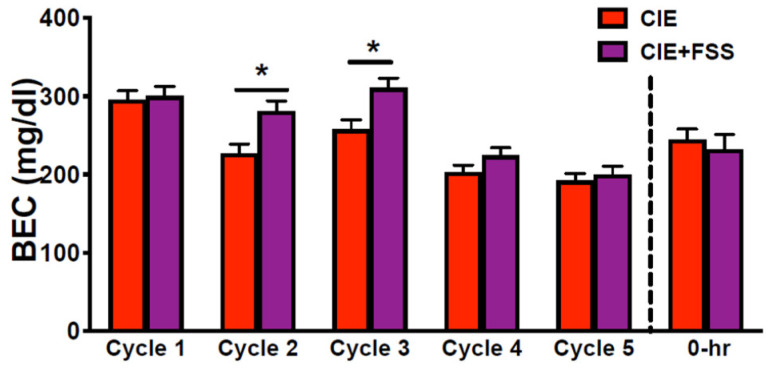
Blood ethanol concentrations (BEC) in mg/dL registered during each CIE exposure cycle and at the 0-h timepoint of tissue collection for the CIE and CIE + FSS groups. BECs were different between these groups only during exposure Cycles 2 and 3 (*).

**Figure 3 brainsci-10-00275-f003:**
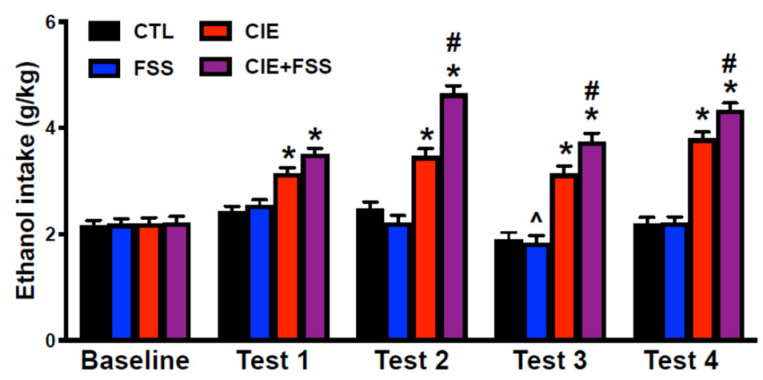
Ethanol intake in g/kg during the baseline and each test cycle after CIE or air exposure for the CTL, FSS, CIE, and CIE + FSS groups. CIE and CIE + FSS mice showed a significantly higher level of voluntary ethanol intake when compared to CTL mice and their own baseline (*). Mice in the FSS group drank less ethanol during Test 3 compared to their own baseline. Mice in the CIE + FSS group consumed the most ethanol compared to all the other groups during Test Cycles 2–4 (#).

**Figure 4 brainsci-10-00275-f004:**
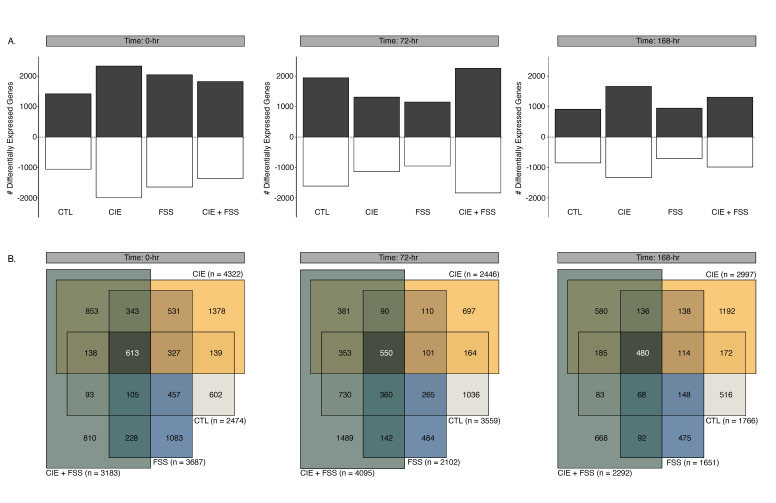
Summary of differential gene expression results for the medial prefrontal cortex (CTX) based on comparison to naïve subjects. (**A**) Bar plots of the number of differentially expressed genes at timepoints 0 h, 72 h, and 168 h. Increased gene expression shown in dark grey, and decreased gene expression shown in white. (**B**) Venn-diagrams showing the four-way overlap of all differentially expressed genes for timepoints 0 h, 72 h, and 168 h. Each of the respective behavioral groups are indicated by a different color: CTL = light grey, CIE = yellow, FSS = blue, and CIE + FSS = green.

**Figure 5 brainsci-10-00275-f005:**
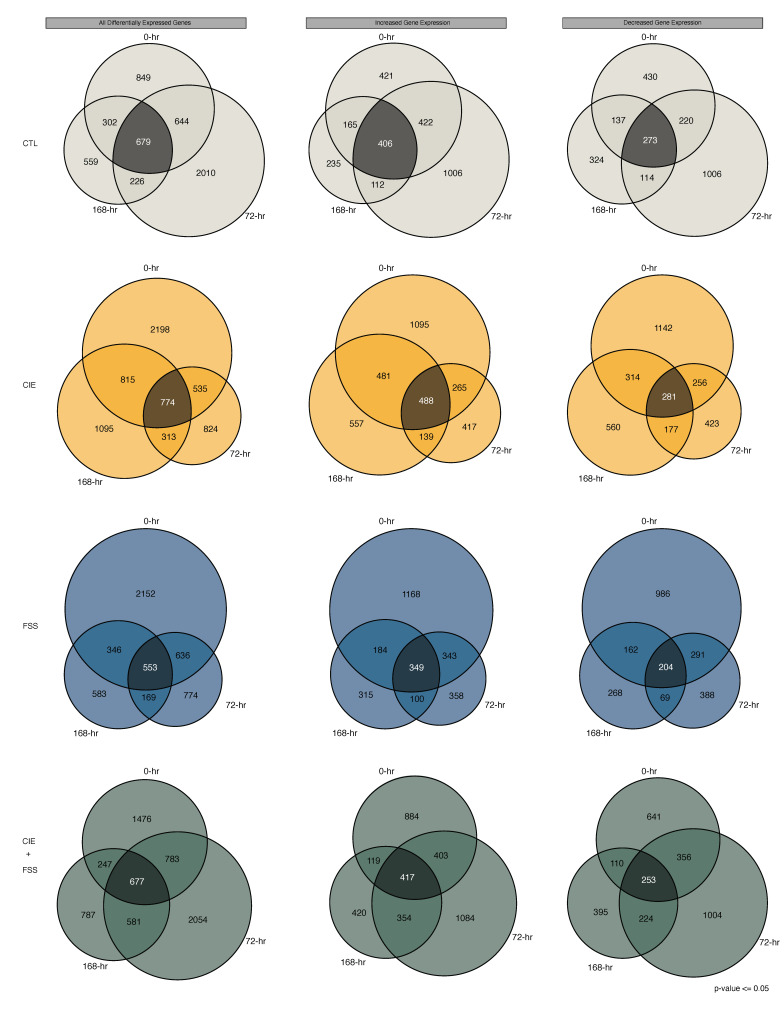
Venn-diagrams within each behavioral group across the examined time-course. Shown are the results for all differentially expressed genes (left), increased gene expression (middle), and decreased gene expression (right) for each of the respective behavioral groups. CTL = light grey, CIE = yellow, FSS = blue, ands CIE + FSS = green.

**Figure 6 brainsci-10-00275-f006:**
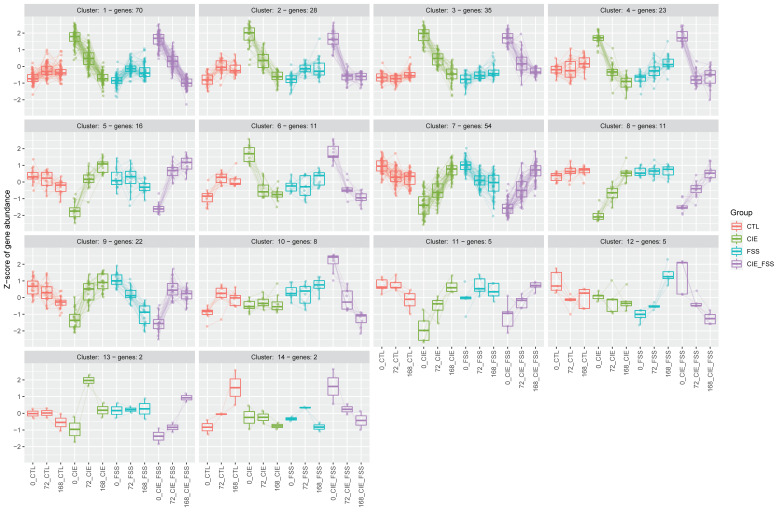
Cluster analysis of gene expressions to define temporal and group-wise differences. Cluster analysis of 292 differentially expressed genes (padj < 0.05) to define temporal and group-wise differences. These clusters were produced using the degPatterns function on the regularized log transformation of the normalized counts. Each cluster represents gene expression patterns shared among differentially expressed genes. Box plots are shown for the expression patterns of each gene within the group. Circles represent the number of genes in each group.

**Table 1 brainsci-10-00275-t001:** Gene ontology analysis of molecular function for (A) increased and (B) decreased differential gene expression, showing canonical groups that are consistently altered across time and uniquely regulated within each behavioral group (CTL, CIE, FSS, and CIE + FSS). The number of genes represent the union of significant genes in that behavioral group and canonical molecular function across time. Results are significantly overrepresented (*p* < 0.05), uniquely with respect to an individual group and timepoint. CTL: non-dependent alcohol consumption, CIE: chronic intermittent ethanol, and FSS: forced stress swim.

Group	Name	000-h	072-h	168-h	Number of Genes
**(A)**					
AIR	RNA polymerase II regulatory region sequence-specific DNA binding (GO:0000977)	5.09E−05	4.86E−02	1.39E−03	89
AIR	kinase binding (GO:0019900)	3.60E−04	1.99E−02	3.50E−03	84
AIR	RNA polymerase II regulatory region DNA binding (GO:0001012)	4.78E−04	2.16E−02	2.48E−03	45
AIR	SUMO binding (GO:0032183)	9.12E−04	2.34E−02	1.55E−02	6
AIR	phosphoprotein phosphatase activity (GO:0004721)	1.55E−03	6.64E−03	4.21E−03	33
AIR	phosphatidylinositol phosphate kinase activity (GO:0016307)	2.89E−03	1.13E−02	4.01E−03	7
AIR	protein kinase binding (GO:0019901)	4.57E−03	5.48E−03	1.33E−02	99
AIR	purine ribonucleoside triphosphate binding (GO:0035639)	1.51E−02	7.31E−03	2.75E−02	69
AIR	ligand-dependent nuclear receptor transcription coactivator activity (GO:0030374)	1.70E−02	2.42E−02	1.04E−02	17
AIR	syntaxin-1 binding (GO:0017075)	1.81E−02	1.94E−03	2.90E−02	6
CIE	phosphatidylinositol binding (GO:0035091)	4.66E−04	3.22E−02	9.13E−03	30
FSS	histone methyltransferase activity (GO:0042054)	7.13E−05	8.36E−05	1.98E−03	18
FSS	steroid hormone receptor binding (GO:0035258)	2.11E−04	4.14E−04	9.74E−04	22
FSS	protein phosphorylated amino acid binding (GO:0045309)	5.05E−04	4.83E−02	2.36E−02	12
FSS	core promoter binding (GO:0001047)	4.51E−03	1.41E−03	2.29E−03	24
FSS	protein tyrosine kinase activity (GO:0004713)	8.19E−03	5.03E−03	4.90E−02	36
FSS	androgen receptor binding (GO:0050681)	9.99E−03	1.28E−04	3.80E−03	11
FSS	regulatory region DNA binding (GO:0000975)	2.16E−02	1.99E−02	2.14E−02	44
FSS	methylated histone binding (GO:0035064)	2.46E−02	4.06E−02	4.66E−03	16
FSS	histone deacetylase binding (GO:0042826)	2.71E−02	3.67E−03	2.05E−02	19
FSS	cadherin binding (GO:0045296)	2.90E−02	2.44E−02	2.62E−02	60
CIE + FSS	transmitter-gated ion channel activity (GO:0022824)	3.11E−02	3.23E−02	2.54E−04	12
**(B)**					
AIR	ubiquitin-specific protease binding (GO:1990381)	4.96E−02	3.50E−02	2.81E−02	6
CIE	GTP binding (GO:0005525)	7.39E−06	9.18E−07	1.96E−03	43
CIE	purine ribonucleoside binding (GO:0032550)	1.61E−05	3.97E−06	9.53E−04	45
CIE	guanyl ribonucleotide binding (GO:0032561)	6.76E−05	1.31E−05	1.00E−03	46
FSS	translation initiation factor activity (GO:0003743)	2.51E−03	4.50E−05	4.00E−02	11
FSS	translation factor activity, RNA binding (GO:0008135)	3.24E−03	7.82E−05	3.21E−02	15
CIE + FSS	syntaxin binding (GO:0019905)	6.20E−03	1.99E−04	1.40E−02	22
CIE + FSS	ubiquitin-protein transferase activity (GO:0004842)	1.83E−02	6.60E−03	2.46E−02	73

**Table 2 brainsci-10-00275-t002:** CNS cell-type analysis of differential expressed genes for CTL, CIE, FSS, and CIE + FSS at time 0 h, 72 h, and 168 h. The four major CNS cell types (neurons, oligodendrocytes, astrocytes, and microglia) represented by genes with 10-fold expression were evaluated for the overrepresentation of (A) increased and (B) differentially expressed genes. The number of genes represent the union of significant genes determined for each behavioral group across the 0-h, 72-h, and 168-h collected timepoints.

Group	Cell-Type	000-h	072-h	168-h	Number of Genes
**(A)**					
	Neurons	1.22E−06	7.53E−23	6.91E−03	113
	Oligodendrocytes	5.49E−01	7.23E−01	6.99E−01	3
AIR	Astrocytes	3.94E−01	2.00E−03	8.32E−01	34
	Microglia	9.79E−01	9.97E−01	4.08E−01	48
	Neurons	7.06E−08	7.68E−03	5.53E−14	106
CIE	Oligodendrocytes	3.52E−01	8.21E−01	6.39E−01	5
	Astrocytes	9.33E−05	2.36E−02	5.06E−01	49
	Microglia	1.87E−20	8.67E−06	3.22E−01	140
	Neurons	1.25E−32	4.23E−06	1.13E−04	128
FSS	Oligodendrocytes	2.68E−01	7.79E−01	1.88E−04	8
	Astrocytes	1.02E−01	1.41E−03	4.36E−01	33
	Microglia	1.00E+00	7.12E−01	8.16E−01	41
	Neurons	2.58E−18	2.17E−08	1.02E−10	141
CIE + FSS	Oligodendrocytes	2.81E−02	7.87E−10	8.68E−02	21
	Astrocytes	2.57E−05	2.13E−05	8.76E−01	59
	Microglia	2.48E−17	2.11E−01	7.06E−01	124
**(B)**					
	Neurons	3.15E−02	5.73E−02	6.22E−02	55
	Oligodendrocytes	7.50E−01	9.08E−04	1.00E+00	9
AIR	Astrocytes	3.36E−03	3.48E−01	1.38E−02	33
	Microglia	9.49E−01	9.25E−01	8.35E−01	42
	Neurons	2.00E−05	7.31E−12	1.61E−03	90
CIE	Oligodendrocytes	2.53E−01	5.76E−02	5.16E−01	7
	Astrocytes	1.42E−11	2.95E−03	1.42E−03	58
	Microglia	9.71E−01	9.74E−01	9.93E−01	46
	Neurons	7.33E−03	1.90E−02	4.62E−02	55
FSS	Oligodendrocytes	8.86E−01	3.48E−01	6.00E−01	4
	Astrocytes	7.93E−02	9.34E−01	9.11E−02	26
	Microglia	7.75E−01	9.99E−01	5.56E−02	46
	Neurons	1.69E−07	2.81E−10	7.85E−03	80
CIE + FSS	Oligodendrocytes	9.64E−02	1.00E+00	1.00E+00	4
	Astrocytes	2.31E−13	5.40E−01	2.16E−06	59
	Microglia	7.69E−01	9.99E−01	1.96E−01	46

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
