# Peer review of "Transcriptome Analysis of Alcohol Drinking in Non-Dependent and Dependent Mice Following Repeated Cycles of Forced Swim Stress Exposure"

_brainsci, 2020, doi:10.3390/brainsci10050275_

Round 1

Reviewer 1 Report

Review:

This manuscript by Farris and colleagues provides a transcriptome analysis of medial prefrontal cortex from male C57BL/6 mice with non-dependent drinking, after CIE dependent alcohol consumption, repeated forces swim stress (FSS), and CIE with repeated FSS.  These studies are important as repeated swim stress is known to promote escalated alcohol consumption in dependent animals, consistent with the known role of chronic stress as a contributing factor to the development of alcohol addiction. Previous studies have shown changes in gene expression with CIE in a variety of brain regions, including prefrontal cortex (ref 13).  However, this study extends that work to examine the combined effect of FSS and CIE on changes in gene expression.  The authors hypothesize that the escalation of alcohol consumption in dependent mice via FSS suggests convergent biological pathways activated by chronic stress and alcohol. 

The paper is well written and contains a large amount of interesting data. These data will be very interesting to the alcohol community and should be made freely accessible. However, there are a number of concerns that should be addressed in the text.  They are listed below.

Issues to be addressed:

  1. While the authors identify the number of animals per group, it is not clear how the RNA samples were prepared. Were mPFC punches collected from both sides of the brain and pooled for each mouse?  Was each mouse a separate sample or were mPFC samples pooled from several animals to generate multiple samples per treatment and time point.  This is not clearly described in the methods and it is not clear how many samples were subjected to RNA-seq for each group at each timepoint.
  2. To understand the depth of sequencing, it would be good to know how many transcripts were detected in each sample.
  3. There is a confusing mix of nomenclature. Throughout the methods section, the AIR group is referred to as CTL.  (Some of the lines are listed here: Lines 88, 90, 97, 108, 123  figure 1 (figure and legend) and figure 3 (figure and legend)).  To be more consistent across the manuscript, it seems that CTL in the methods section should be converted to AIR, as there appears to also be a naïve control group that was used for all the comparisons.  In addition, the CTL(AIR) group (no CIE, no FSS, but non-dependent drinking) is referred to as AIR in the results and discussion and abstract.  It is important to be consistent throughout the manuscript and the tables and Figure 4-6 all use AIR.
  4. The authors should be clearer in their description of the analysis for differential gene expression. Lines 157-158 state that all treatment groups were compared to naïve control=AIR.  Yet lines 214-215 state that expression changes were determined for each group (including AIR) by comparison to naïve controls.  This seems to be a much more appropriate description to this reviewer.  Please clarify in text so that this is clear.
  5. Pyrazole was administered to all 4 treatment groups, but not naïve controls, so how does that affect the results (could there be similar gene expression changes in all groups from pyrazole?). On line 233, the authors discuss 204-488 gene that overlap across treatment groups- could pyrazole contribute to that? The authors may want to discuss this further in the results section.
  6. Line 113- mentions females, but no females included in the study
  7. Potentially change title as data include non-dependent mice as well
  8. It would have been useful to have a bit more discussion of the similarities of the top 500 genes reported in Ref 13 to the most highly differentially expressed genes in this study in the CIE group. Were data largely consistent across the two methods even though arrays tested fewer genes?
  9. It would be most useful if the full data set is uploaded to GEO or some other RNA-seq repository (or all fold changes listed in excel files in supplementary data (as done in Ref 13 for top 500 differentially expressed genes). Will the full data set be released upon publication as encouraged by NIH guidelines?

Reviewer 2 Report

This manuscript describes a well-designed study to explore the interaction between chronic intermittent ethanol exposure and chronic stress induced by a forced swim paradigm on transcriptomic changes in the prefrontal cortex. This is both an interesting and relevant question. My major concerns about the paper focus on the approach for the statistical analysis and the lack of clarity in description of methods/results from some analyses. Overall, this is sound study and my concerns should be easy to address.

Major Concerns

  • One of my major concerns is the analysis approach for the voluntary alcohol consumption and the differential expression study. I appreciate that this is a fairly complex study design with many questions/comparisons of interests. For the voluntary consumption, the authors used a two-way ANOVA with group and phase rather than a three-way ANOVA that would have separated group into FSS and CIE effects. In the authors’ version, many group-wise comparisons were made but no higher order ‘effect’ comparisons were made. For example, they could have compared the effect of CIE in the group that did not receive the FSS to the group that did receive FSS at a given phase. In the case of voluntary consumption, the results were likely to be similar to what as actually calculated CIE vs. CIE-FSS at a time point, since values were similar between CTL and FSS, so I am not as concerned about this particular analysis. For the DE study, the authors took the approach of comparing all groups and time points to a control and then identifying overlaps in significance. A more statistically rigorous approach would be to use a three-way ANOVA (time, stress, CIE) and to specifically examine the interaction effects, i.e., an interaction between stress and CIE. This way you can test direct the difference in the effect of CIE between the group that was exposed to the forced swim stress vs. the group that was not.

Minor Issues

  • It isn’t clear whether both male and female mice were included in the analysis. In the ‘Animal Model’ section of the Materials and Methods (line79), the authors indicate that only male mice were purchased, but in the ‘CIE Exposure’ section of Materials and Methods (line 113), the authors indicate that separate inhalation chambers were used for males and females. If females were included, sex as a covariate needs to be addressed throughout.
  • Figure 1 – the caption does not indicate what the red triangles represent. It would also be helpful in this figure to indicate that a stable baseline was established over 3 weeks. With cursory glance at the figure, it appears that mice were only acclimated to the alcohol consumption paradigm over five days.
  • Were any outlier mitigation techniques used for the daily intake values prior to taking a mean value within each test cycle?
  • What software and what package/procedure was used for the repeated measures ANOVA for the behaviors/BEC?
  • What was the rationale for not using a multiple testing correction in the DGE analysis (Line 158)?
  • There is some contradiction about what group was used for the ‘naïve controls’. On line 132, the authors indicate that there was a naïve control group. Was this group not exposed to alcohol consumption paradigm, the CIE, or the FSS? On lines 157 and 158, the authors indicate that the naïve controls are the AIR subjects but later describe differences between the AIR group and the control.
  • The authors over-interpreted the main effect of group in the presence of an interaction effect in BEC (Lines 176-180).
  • Why was time of brain collection included in the voluntary ethanol intake initially? Was it removed before formally testing for group and phase?
  • Please include general alignment and quality results for the Tag-Seq data along with precise sample sizes
  • The sentence that starts on line 215, “Each of the four treatment groups received intraperitoneal injections of pyrazole to inhibit alcohol metabolism and maintain consistent blood ethanol concentrations”, is a bit vague. When were these injections given? How is this relevant to the DE analysis?
  • For the paragraph starting at line 223, it would be helpful to reiterate what group all other groups were compared to for differential expression. What are the units for the effect sizes indicated in this paragraph? Log 2 fold change?
  • The sentence that starts at line 229, “Temporal patterns of gene expression….” is unclear.
  • In line 236, the authors compare their genes to previous research. It would be helpful here to not just get the number that overlapped but the total number identified as associated with CIE in both studies to understand the percent of overlap.
  • Figure 4 – increase font size; include in figure legend what comparison was tested, i.e., what was the control group that these were differentially expressed in comparison to.
  • For the paragraph that starts on line 253, I wasn’t able to find any reference to miR-206 in the supplementary table 1. Was the comparison of targets done as part of the EnrichR analysis or was this done using a different software?
  • What was the rational for doing the enrichment separately for up-regulated and down-regulated genes? For many of the canonical gene ontologies and molecular pathways, direction is not relevant, e.g., a pathway may include both activators and repressors.
  • Table 1 – please include the statistical requirements for genes to be included in the enrichment analysis and the statistical requirements for a term being included in the table within the table legend; it isn’t clear what any of the number in the table represent; could this have been better represented by a heatmap or some other figure?
  • In the paragraph describing the clustering results (starting at line 292), it would be helpful to have the number of genes that were included in this analysis stated early on.
  • Line 297, how do you know that cluster 1 is the most robust?
  • Figure 6 – need more detail in the figure legend, e.g., what do the circles represent; Can we assume that padj stand for false discovery rate? Were the controls not included assuming there was a control group with no alcohol exposure/air chamber exposure?
  • Line 332-333, This, “multiple systems were differentially affected with respect to their environmental influence”, was not directly tested but implied by one being significant and the other not. It could be tested though.
  • Table 2 – it isn’t clear what the numbers in this table represent; the table legend does not indicate the difference between A and B; if B represents down regulated genes, what is the biological interpretation, a lack of that cell type?

Reviewer 3 Report

I've attached a Word document with tracked changes

Round 2

Reviewer 2 Report

The reviews have addressed most of my comments and suggestions. Two minor issues remain that can easily be addressed:

1) The authors have still not indicated what 'padj' stands for in the text or table.

2) The microRNA results from EnrichR are not in Supplementary Table 1. I am  not seeing results for mir-206 as indicated in the text.

Author Response

Here I address the two remaining issues that require clarification. 1. "padj" - this refers to an adjusted p value based on the false discovery rate (FDR). 2. Suppl Table 1 - the reviewer is correct, I apparently sent the old version of this table. The revised one that includes microRNA results is attached. I hope this addresses all remaining issues regarding our manuscript.